# Effect of Brown Algae and Lichen Extracts on the SCOBY Microbiome and Kombucha Properties

**DOI:** 10.3390/foods12010047

**Published:** 2022-12-22

**Authors:** Darya A. Golovkina, Elena V. Zhurishkina, Olga N. Ayrapetyan, Artem E. Komissarov, Anastasiya S. Krylova, Elizaveta N. Vinogradova, Stepan V. Toshchakov, Filipp K. Ermilov, Artak M. Barsegyan, Anna A. Kulminskaya, Irina M. Lapina

**Affiliations:** 1Petersburg Nuclear Physics Institute Named by B.P. Konstantinov of National Research Center “Kurchatov Institute”, Mkr. Orlova Roshcha, 1, 188300 Gatchina, Russia; 2Kurchatov Genome Center—PNPI, Mkr. Orlova Roshcha, 1, 188300 Gatchina, Russia; 3Center for Genome Research NRC “Kurchatov Institute”, Kurchatov Sq. 1, 123098 Moscow, Russia

**Keywords:** kombucha, *Fucus vesiculosus*, *Cetraria islandica*, sugar content, antioxidant activity, metagenomic analysis, nutraceutical, synbiotics

## Abstract

Kombucha tea was made by the fermentation of SCOBY culture of green tea broth with the addition of *Fucus vesiculosus* algae extract, *Cetraria islandica* lichen extract and their mixture. Kombucha was also made without the herbal supplements as a control. After 11 days of fermentation, in addition to the yeast *Brettanomyces bruxellensis* and the bacteria *Komagataeibacter rhaeticus* and *Komagataeibacter hansenii* contained in all of the samples, the yeast *Zygosaccharomyces bailii* and bacteria *Komagataeibacter cocois* were detected in the samples with the herbal extracts. In all of the kombucha with herbal additives, the total fraction of yeast was decreased as compared to the control. The total content of polyphenols and the antioxidant activity of the beverages with and without the addition of herbal extracts were comparable. The kombucha made with the algae extract showed an increased content of sucrose and organic acids, while the fructose and glucose content in the samples with algae and the mixture of extracts were lower than in the other samples. The samples with the algae extract had the highest organoleptic indicators “aroma”, “clarity” and “acidity”, while the control samples had slightly higher indicators of “taste” and “aftertaste”. The results of this study indicate the potential of algae and lichens as functional supplements for obtaining non-alcoholic fermented beverages with additional nutraceutical value.

## 1. Introduction

Striving for a healthy lifestyle involves enriching the diet with healthy foods. Kombucha is a healthy drink, resulting from tea fermentation performed by a special starter including bacteria and yeast called SCOBY (Symbiotic Culture of Bacteria and Yeast). The SCOBY is added to freshly prepared tea broth as a cellulose film or fermentation broth and then incubated for 7–14 days to get the fermented drink [1,2]. The characteristic microbiome of kombucha includes yeast and acetic acid bacteria, which interact through co-metabolism to promote the synthesis and organoleptic qualities of the final beverage [3]. During the fermentation of kombucha, metabolic reactions occur leading to the formation of intermediate products and secondary metabolites. The production of the compounds is directly related to the specific microbiota and cross-feeding between them. Beyond the microorganisms forming a symbiotic culture, the factors that will determine the growth and activity of the kombucha consortia include the substrate, carbon sources used and fermentation parameters. As a result, the fermented teas can have vastly different properties and chemical compositions.

To obtain a range of kombucha-based fermented drinks and to give them additional beneficial properties, both alternative raw materials for kombucha fermentation [4,5] and a variety of tea additives [6] are actively used. Promising components for functional foods include brown algae extracts containing a number of biologically active compounds, such as polysaccharides, polyphenolic compounds, proteins, minerals, iodine, vitamins and fatty acids [7,8,9,10]. In general, the brown algae components are characterized as having immunomodulatory, antimicrobial and antitumor activities [11,12,13]. The extracts from lichens traditionally used in folk medicine are also of particular interest [14]. The main components of the composition of the aqueous extract of Iceland moss (*Cetraria islandica*) are beta-glucans, organic acids and trace elements. As shown in a number of studies, the aqueous and ethanol extracts of lichens exhibit immunomodulatory, antimicrobial and anti-inflammatory activities, as well as reducing the damage to the body caused by diabetes [15,16].

When using new substrates for the fermentation of kombucha, new chemical compounds are introduced into the culture medium, the activity of which against SCOBY is not fully known [1]. Moreover, research data on the selection of the strains of microorganisms present in SCOBY and their adaptation to a new environment enriched with materials other than teas is limited, and represent the prospect of further research.

In this study, we analyzed the effect of additives such as *Fucus vesiculosus* brown algae extract, *Cetraria islandica* lichen extract and mixtures of these extracts on the microbiome of the SCOBY symbiotic culture, and on the composition and properties of the resulting drinks.

## 2. Materials and Methods

### 2.1. Plants and Chemicals

All reagents were purchased from Sigma-Aldrich (Darmstadt, Germany) and Acros Organics (Branchburg, NJ, USA) unless otherwise specified.

The thalli of brown algae *F. vesiculosus* were collected in the Zelenetskaya Bay of the Barents Sea (WSG84 N69.116, E36.089). Seaweeds were identified by Dr. E. Obluchinskaya and the voucher specimen was deposited in the Collection of the Zoobenthos Laboratory (Murmansk Marine Biological Institute of the Russian Academy of Sciences). The algae were air dried at 20 °C.

The thallus of the lichen *C. islandica* was collected in pine forests in the Lakhdenpokh region of Karelia (WSG84 N61.589, E31.488) and identified by Dr. T. Danchul (Komarov Botanical Institute of the Russian Academy of Sciences). The lichens were air dried at 20 °C.

### 2.2. Preparation of Experimental Set Ups

#### 2.2.1. Preparation of Medium for Culturing SCOBY

Green leaf tea (Ahmad^TM^, Moscow, Russia) was added to boiling tap water (100 mL of water per 2 g of dry tea), and 8% (*w*/*v*) sucrose was added. After cooling to room temperature, the tea leaves were removed from the solution by straining.

#### 2.2.2. Algae Extract Preparation

Dry ground algae, *F. vesiculosus* (7 g), was added to distilled water (140 mL) and stirred on a magnetic stirrer at 95 °C for 2 h. The extract was left to infuse at 20 °C and the next day was centrifuged at 3000 rpm (Eppendorf centrifuge 5810R). Next, the supernatant was used to prepare the culture medium.

#### 2.2.3. Preparation of Lichen Extract

Dry ground lichen, *C. islandica* (10 g), was added to distilled water (300 mL), preheated to 95 °C and stirred on a magnetic stirrer at 95 °C for 1 h. The extract was left to infuse at 20 °C and the next day was centrifuged at 3000 rpm/min (Eppendorf centrifuge 5810 R). Next, the supernatant was used to prepare the culture medium.

### 2.3. Cultivation of SCOBY Specimens

The SCOBY starter culture was purchased from OOO “Taliya” (Moscow, Russia). Before being introduced into the experiment, symbiotic microorganisms were cultivated at 27 °C for 12 days in green tea with sucrose (item 2.1). The liquid culture thus obtained was further cultivated in the following combinations:

K: green tea with sucrose + 10% (*v*/*v*) SCOBY liquid culture;

KF: green tea with sucrose + 10% (*v*/*v*) SCOBY liquid culture + 3% (*v*/*v*) algae extract;

KC: green tea with sucrose + 10% (*v*/*v*) SCOBY liquid culture + 3% (*v*/*v*) lichen extract;

KFC: green tea with sucrose + 10% (*v*/*v*) SCOBY liquid culture + 3% (*v*/*v*) algae extract + 3% (*v*/*v*) lichen extract.

Each of the samples was cultivated for 14 days at a temperature of 27 °C.

Aliquot samples from 3 replications of the fermentation batch were aseptically collected at the time points of 0, 7, 11 and 14 days of fermentation for the determination of pH, total polyphenolic content and antioxidant activity.

To determine the parameters of the fermented kombucha tea (chemical, metagenomic and organoleptic analyses), the samples were taken on the 12th day of fermentation.

### 2.4. Determination of Parameters of Kombucha Fermentation

#### 2.4.1. pH Determination

The pH of kombucha samples was measured using an electronic pH meter HI 8519 (Hanna Instruments Inc., Woonsocket, RI, USA).

#### 2.4.2. Determination of Total Polyphenolic Content (TPC)

The total polyphenolic content (TPC) compounds in the kombucha samples were quantified using the Folin–Ciocâlteu method as described in [17] with slight modifications. The kombucha samples were diluted with distilled water in a 20-fold dilution. A total of 100 μL of the Folin–Ciocâlteu reagent and 2 mL of distilled water were added to the prepared diluted kombucha sample (100 μL), and after 5 min of staying 1 mL of 20% Na_2_CO_3_ was added. After storing for 1 h at 20 °C in the dark, the absorbance of the mixture was measured at 765 nm on a JASCO V 560 spectrophotometer (JASCO Corporation, Tokyo, Japan). Phloroglucinol was used as a standard in order to plot the calibration curve and the total polyphenolic content (TPC) was expressed as mg of phloroglucinol equivalents per ml of kombucha.

#### 2.4.3. Determination of Antioxidant Activity

Antioxidant activity of kombucha was evaluated using Blois method for DPPH radical scavenging assay as described in [18] with slight modifications. Briefly, 0.5 mL of kombucha sample was diluted with 1.5 mL 95% CH_3_OH and then 2 mL of 1 mM DPPH solution in methanol was added to the sample. The reaction mixture was then incubated in the dark for 30 min at 20 °C. Control samples containing 1 mL 95% CH_3_OH + 1 mL 1 mM DPPH were incubated under the same conditions. The absorbance was read at 517 nm on a JASCO V 560 spectrophotometer (JASCO Corporation, Tokyo, Japan). The antioxidant activity was measured as a percentage of free radical inhibition, according to the formula: %inhibition = [(absorbance of the control − absorbance of the sample)/absorbance of the control)] × 100%. An additional control experiment was performed following the same procedure using ascorbic acid as a standard.

### 2.5. Determining the Parameters of Kombucha Tea

#### 2.5.1. Chemical Analysis of the Composition of Extracts of Tea, Algae, Lichen and Kombucha Samples

To carry out chemical analysis, the samples of the liquid phase of kombucha on the 0th and 12th day of fermentation and extracts of tea, algae and lichen were centrifuged at 5000 rpm and 4 °C cooling (Eppendorf centrifuge 5810 R). The supernatant was dried on a freeze dryer Martin Christ Alpha 1-2 LD plus (Martin Christ Gefriertrocknungsanlagen GmbH, Osterode am Harz, Germany).

The carbohydrate, organic acids and other metabolite content of tea extracts, algae, lichen and samples of kombucha were quantified with gas chromatography of trimethylsilyl derivatives according to [19]. The samples were treated with 1,1,1,3,3,3-hexamethyldisilazane in a mixture of 1 mL of pyridine and 1 mL of acetonitrile in the presence of trifluoroacetic acid at 60 °C for 1 h. The resulting solution was placed into the sampler of the chromatograph. Analysis conditions: SBP5-25 column (25 m × 0.25 mm × 0.2 μm); carrier gas N_2_, 20 cm/s; temperature program: 1 min at 70 °C, rise 4 °C/min up to 320 °C, 5 min at 320 °C; sample injection temperature 240 °C, flow divider 1:20, sample volume 2 µL; flame ionization detector, temperature 325 °C, hydrogen supply rate: 40 mL/min, nitrogen: 25 mL/min, oxygen: 250 mL/min. Peaks were assigned according to retention times after a series of calibration analyses of model mixtures of a given composition.

The acetic and formic acids were determined using HPLC according to [20]. The samples were analyzed in LC-10A chromatograph equipped with PAD at 210 nm (Shimadzu Corp., Kyoto, Japan). Samples were injected into a glass column 9 × 500 mm with resin 2614 (Hitachi, Ltd., Tokyo, Japan) in H+-form at 55 °C and eluted with 10 mM perchloric acid (1.2 mL/min).

#### 2.5.2. DNA Extraction, Preparation and Sequencing of Shotgun Metagenomic Libraries

To carry out metagenomic analysis of the microbial composition, kombucha liquid phase samples were taken on the 12th day of fermentation and then were centrifuged at 5000 rpm and 4 °C cooling (Beckman Coulter Avanti J25i). The supernatant was removed, and the precipitate was kept under buffer (150 mM NaCl, 100 mM EDTA, 100 mM Tris-HCl (pH = 8.0)) at a temperature of 4–8 °C until the study.

The DNA extraction from microbial consortia was performed using DNeasy PowerLyzer Microbial kit (Qiagen, Hilden, Germany), using the manufacturer’s instructions. The concentration was assessed using Qubit fluorimeter (Thermo Fischer Scientific, Waltham, MA, USA) and approximately 50 ng of DNA was used for library preparation. Libraries were prepared using Illumina DNA Prep kit according to the manufacturer’s instructions. The quality of libraries was assessed using capillary electrophoresis on Bioanalyzer 2100 instrument (Agilent Technologies, Santa Clara, CA, USA). Final libraries were sequenced on Novaseq 6000 instrument using 2 × 150 bp paired-end read chemistry.

#### 2.5.3. Bioinformatic Analysis

The sequencing results were analyzed in several steps. The first step was to check the quality and validity of data using the FASTQC software (https://www.bioinformatics.babraham.ac.uk/projects/fastqc/, accessed on 15 July 2022) and Trimmomatic (https://github.com/usadellab/trimmomatic, accessed on 15 July 2022). Genome-wide assembly was then carried out using the metaSPAdes software [21]. Then, using the metaWRAP software [22], long contigs were assembled. Each contig was then identified based on the NCBI BLAST database.

#### 2.5.4. Sensory Evaluation

Sensory evaluation was carried out on the 12th day of fermentation of kombucha and included taste, flavor, aftertaste, clarity and acidity of beverages. Each untrained participant (n = 18, age from 18 to 52) was served 50 mL of kombucha from each sample. Panelists were required to indicate their preferences among kombucha samples based on a five-point hedonic scale (5: very good, 4: good, 3: middle, 2: bad, 1: very bad) and the final results were interpreted based on the average score of each parameter [23]. Samples were presented in random order and identified with random three-digit codes. To neutralize the taste, drinking water was served between the samples.

### 2.6. Statistical Processing Analysis

For statistical data processing, plotting charts and diagrams Excel 2010 (Microsoft) and OriginPro 16 (Microcalc) were used. The graphs and tables show average values of at least 3 independent experiments, bars represent the standard errors. The statistical significance of the observed differences was assessed using Student’s t-test.

## 3. Results

The addition of extracts in a volume of 3% of the composition of the culture medium preceded the preliminary organoleptic and microbiological experiments (the data are not shown). On the one hand, we sought the pleasant taste of the drinks with the plant extracts. On the other hand, the addition of the algae extracts above 3% of the volume of the culture medium led to a marked suppression of the growth of the SCOBY culture. The maturity of the drinks on the 12th day of fermentation was determined on the basis of the organoleptic tests. The chemical composition of the extracts of green tea, brown algae *Fucus vesiculosus* and lichen *Cetraria islandica* are shown in (Appendix A).

### 3.1. Parameters Measured during Kombucha Fermentation

#### 3.1.1. pH

The presence of algae extracts in the culture medium lowered the pH of the KF and KFC samples as compared to the K sample from the 4th day of fermentation until the end of the experiment (Figure 1). The addition of the lichen extract alone (KC sample) had practically no effect on the pH of the drink during fermentation compared to the K sample.

#### 3.1.2. Total Polyphenolic Content (TPC)

The addition of the algae and lichen extracts into the culture medium reduced (from 3.6 ± 0.05 to 2.74 ± 0.2 mg/mL) the TPC in the KFC sample compared to the control only at the start of fermentation (Figure 2). There was no significant difference in the TPC levels between the samples with the herbal extracts compared to the control on subsequent days of fermentation.

#### 3.1.3. Antioxidant Activity

The antioxidant activity of the drinks with the plant extracts was decreased on day 0 (KF sample by 3% and KFC sample by 5.8%), on day 4 (KF sample by 3.1%, KC sample by 2.75% and KFC sample by 4.3%) and on day 7 (KC sample by 2.6%) of fermentation as compared to the control. However, as the drinks approached maturity (after 11 days of fermentation), the antioxidant activity of the drinks with the plant extracts was comparable to the activity of the control (Figure 3). Continuing the fermentation after the drinks reached maturity resulted in the decrease in the antioxidant activity in all of the kombucha.

### 3.2. Parameters of Kombucha Tea

#### 3.2.1. Chemical Analysis

Sugar Content

When analyzing the carbohydrate content of the kombucha samples, it was revealed that the sucrose content decreased by 6.4% in the control kombucha sample K by the 12th day of fermentation. In the KF sample, by the 12th day of fermentation the sucrose content exceeded that of the other samples (decreased by 3.6% since the start of fermentation). The products of sucrose cleavage, fructose and glucose, were detected in the KF sample in a lower amount than in the other samples (Figure 4b,c; Appendix A). In the KC sample, the least amount of sucrose was contained (decreased by 14.1%), while a high content of fructose and glucose was detected, similar to the control. In the KFC sample, an average sucrose content was detected (decreased by 8.2% since the start of fermentation) relative to the other samples. However, the content of fructose and glucose was below average.

Organic Acids

Ten types of organic acids were detected in the kombucha samples at 0 and 12 days of fermentation (Figure 5). By the 12th day of fermentation, the total content of organic acids in the KF sample with the addition of the algae extract was greater than in the other samples (12.56 mg/g). For the rest of the samples, the values of this parameter were as follows: 10.04 mg/g for the K sample, 9.74 mg/g for the KC sample and 8.6 mg/g for the KFC sample. The content of acetic acid increased in all of the beverage samples by the 12th day of fermentation. The highest level of acetic acid was detected in the KF sample (8.94 mg/g), while for the K, KS and KFC samples the level was 4.44 mg/g, 5.12 mg/g and 6.68 mg/g, respectively (Figure 5). Interestingly, pyruvic acid (3.65 mg/g) was found only in the kombucha sample K on the 12th day of fermentation. On the 12th day of fermentation, gluconic acid was found in all of the samples of kombucha. The greatest amount of gluconic acid was measured in samples KF (1.85 mg/g) and KFC (1.92 mg/g), while there was only 1.28 mg/g and 1.38 mg/g of gluconic acid in samples K and KC, respectively (Appendix A). Formic acid was detected on the 12th day of fermentation in the KF and KC samples (1.21 and 2.14 mg/g, respectively).

Quinic acid was detected in all of the kombucha samples at the beginning of fermentation. Nevertheless, after 12 days of fermentation, it was not detected in the KF and KFC samples, and its content decreased in the K and KC samples (from 1.75 to 0.67 mg/g for sample K, from 1.84 to 1.10 mg/g for sample KC).

Amino Acids

Alanine was detected in samples K and KC (0.40 and 0.38 mg/g, respectively), measured on the 12th day of fermentation. Glycine was found in all of the fermented drinks except in the KFC sample (0.20 mg/g in the K sample, 0.24 mg/g in the KF sample, 0.15 mg/g in the KC sample). Lysine was detected only at the beginning of fermentation in the KC and KFC samples (0.56 and 1.22 mg/g, respectively).

Other Metabolites

In all of the fermented drinks on the 12th day of fermentation, orotic acid (a vitamin-like substance) was detected, and the largest amount of it was found in the KF sample (1.08 mg/g). For the KFC, K and KC samples, the values were 0.88 mg/g, 0.48 mg/g and 0.23 mg/g, respectively.

Phenolic acids were also detected in the kombucha on the 12th day of fermentation: 4-coumaric acid in the K sample (0.09 mg/g), KF sample (1.50 mg/g) and KFC sample (1.61 mg/g); ferulic acid in the K sample (0.22 mg/g), KC sample (0.24 mg/g) and KFC sample (0.26 mg/g).

Glycerol was found in all of the samples of kombucha on the 12th day of fermentation (0.58 mg/g in the K sample, 1.34 mg/g in the KF sample, 0.47 mg/g in the KC sample and 1.23 mg/g in the KFC sample). At the beginning of fermentation, the polyols were detected: glycerol (0.22 mg/g in the K sample), myo-inositol (0.27 mg/g in the K sample, 0.30 mg/g in the KF sample, 0.29 mg/g in the KC sample and 0.31 mg/g in the KFC sample), and mannitol in the KF and KFC samples (0.57 and 0.50 mg/g, respectively).

#### 3.2.2. Metagenomic Analysis

Shotgun metagenomics were used to examine the microbiota of the kombucha communities after 11 days of fermentation. In general, in the samples with the plant extracts the amount of yeast was reduced compared to the original culture (by 6% in the KF and KC samples, by 7% in the KFC sample) (Figure 6). Moreover, the microbiome of the SCOBY culture became more diverse in the samples containing algae (KF and KFC). All of the detected bacteria in the microbiota of the strains belonged to the genus *Komagataeibacter* of the Acetobacteraceae family, among which the bacteria *Komagataeibacter rhaeticus* predominate (60% in the K sample, 51% in the KF sample, 62% in the KC sample and 72% in the KFC sample), followed by *Komagataeibacter hansenii* (22% in the K sample, 24% in the KF sample, 26% in the KC sample and 12% in the KFC sample). However, the *Komagataeibacter cocois* bacteria were present in the samples of KF and KFC kombucha (13 and 5%, respectively). The yeast *Brettanomyces bruxellensis* was found in all of the samples (18% in the K sample, 9% in the KF sample, 12% in the KC sample and 8% in the KFC sample). However, in addition to *Brettanomyces bruxellensis*, the KF and KFC kombucha samples contained the yeast *Zygosaccharomyces bailii* (3% each). Thus, the *Cetraria islandica* extract (KC sample) had the least impact on the culture microbiome of SCOBY.

#### 3.2.3. Sensory Evaluation

The sensory evaluations of the kombucha samples resulted in the highest values of the organoleptic indicators “flavor”, “clarity” and “acidity” for the KF sample, and the highest values of “taste” and “aftertaste” for the K sample. (Figure 7).

## 4. Discussion

Recently, modern diets increasingly include the refreshing drink “kombucha”, which consumers appreciate for its pleasant, sour taste and healing properties. New studies are constantly emerging that comprehensively study the properties of the finished drink and the parameters of kombucha fermentation, including in-depth microbiological studies of bacteria and yeast using a combination of cultural and genetic methods [17]. Usage of alternative materials as a substrate for SCOBY fermentation is also widely studied. For example, the authors of the review [4] consider the effect of tea supplements (cinnamon, apple juice), infusions of coffee, lemon balm, winter savory, yarrow, fruit juices, milk, and even of byproducts and waste (soy whey and banana peel extract) on the properties and composition of fermented drinks.

The extracts of the brown algae *Fucus vesiculosus* and lichen *Cetraria islandica* can be of additional nutraceutical value in the composition of the drink since they have antitumor, immunomodulatory, antimicrobial and anti-inflammatory activities [9,13,15,16]. To date, there is a small amount of research describing the use of seaweed as an ingredient in the production of a fermented drink. For example, in the article [24], the authors compared the properties of kombucha prepared with the extracts of the red seaweed *Porphyra dentata* with black and green tea kombucha. Permatasari and colleagues investigated the properties of kombucha tea based on the green seaweed *Caulerpa racemosa* as a potential functional beverage to combat obesity and as an anti-ageing food [25,26].

However, the effect of herbal supplements, such as brown algae and lichen extracts, on the composition of fermented tea and on the microbiological and biochemical properties of kombucha have never been studied before. There was no complete replacement of fermented tea with the alternative raw materials in this study, and the additives of the plant extracts were only introduced in the amount of 3% (6% in the case of the KFC sample). Nevertheless, discrepancies were observed both in the characteristics of the fermentation process of the kombucha with the supplements compared to the original kombucha, and in the properties of the finished product.

One of the main valuable qualities of kombucha is its antioxidant properties; that is, its ability to neutralize the oxidative action of free radicals. A number of studies have shown that the antioxidant activity of kombucha is higher than that of non-fermented tea, which is the basis of the drink [18,27]. Moreover, green tea-based kombucha has higher antioxidant activity compared to fermented black tea beverages [17,24,28,29,30]. In some studies, the DPPH scavenging activity of a drink correlated with the content of the polyphenols during fermentation [17]. In a study [18], the antioxidant activity of kombucha decreased with the long-term storage of the drink, although the content of the polyphenols was kept at a relatively constant level. In the studies [24,31], the total content of polyphenols did not correlate with the DPPH scavenging activity. The authors Aung and Eun [24] suggested that the antioxidant activity in their kombucha samples increased, compared to the beginning of fermentation due to the possible synergistic effect of organic acids and the influence of metabolites formed during fermentation. In this study, the content of the polyphenols in all of the kombucha samples did not change significantly from day 4 to 14 of fermentation (Figure 2). The antioxidant activity of the kombucha samples with the addition of the plant extracts from 0 to 7 days of fermentation was lower than that of the original kombucha (K sample) (Figure 3); however, by the 11th day of fermentation, the DPPH scavenging activity of samples KF, KC and KFC reached and was comparable to the level in the sample K. Thus, it can be said that the addition of the algae and lichen extracts to fermented green tea does not adversely affect the level of antioxidant activity in finished fermented drinks.

A study [32] stated that the chemical composition and sensory characteristics of the drink were mainly influenced by the composition of the yeast in the kombucha consortium. Metagenomic analysis revealed that *Brettanomyces bruxellensis* was the predominant yeast culture on the 11th day of fermentation in all of the kombucha samples, and the only yeast in samples K and KC (Figure 6). The predominance of this strain was also observed in the liquid phase of the kombucha after 15 days of fermentation in [3] and in the solid phase (cellulose film) of the kombucha in [33]. Interestingly, in [32,34] *Brettanomyces bruxellensis* was chosen as one of the yeast monocultures for the composition of the minimum set of microorganisms necessary for the production of kombucha. In this experiment, we observed a decrease in the total amount of yeast in all of the samples of kombucha with the addition of the plant extracts. There was also a change in the composition of the microbiota in the KF and KFC samples compared to the initial culture K (Figure 6). It is possible that the addition of the plant extracts to the culture medium created less favorable conditions for yeast growth, since the extracts contained components that have an antimicrobial effect (fucoidans in the algae, usnic acid in the lichen [16,35,36]).

The fermentation of tea from initiation to the finished drink takes place through several enzymatic phases. The complex interactions of the multi-species microbial ecosystem of the kombucha are characterized by both cooperation and competition between microbes in the kombucha solution [37]. Presumably, the reason for the relatively low content of fructose and glucose in the KF and KFC samples on the 11th day of fermentation (Figure 4) was the introduction of the algae extract into the culture medium, which components ability to inhibit the activity of the enzymes [38,39] and prevented the breakdown of sucrose by the microorganisms. It is possible that the introduction of the plant extracts into the culture medium also affected the composition and amounts of enzymes secreted by the SCOBY consortium, which, in turn, could lead to a change in the chemical composition of the drinks, in particular, carbohydrate composition. On the other hand, in the KFC sample on the 12th day of fermentation, the increased content of the acetoacetic bacteria *Komagataeibacter rhaeticus* was detected, exceeding their content in the K sample by 12%, in the KF sample by 21% and in the KC sample by 10% (Figure 6). It is also possible that the reason for the low content of monosaccharides in the KFC sample is their accelerated metabolism by this type of bacteria. In any case, these versions require additional experimental checks.

As known, the decrease in the pH of the drink during the fermentation of the kombucha is due to an increase in the content of organic acids in the culture medium produced by acetic acid bacteria. In addition, low pH supports the growth of acetic acid bacteria and yeasts, which cleaves sucrose [3,29]. In our case, a lower pH value was detected in the KF and KFC samples (*p* < 0.05) compared to the K sample starting from the 4th day of fermentation (Figure 1). Obviously, it is acetic acid that makes the greatest contribution to the acidity increase in the fermented drink. Its content usually prevails among the organic acids in the kombucha [17,29]. The acetic acid level was higher in the KF and KFC samples (8.94 mg/g and 6.68 mg/g, respectively) (Appendix A) on the 12th day of fermentation. These data also correspond to the results of the organoleptic analysis, according to which the samples of the finished drinks in the KF and KFC samples had the highest acidity (Figure 7).

The organoleptic properties of kombucha are affected both by the enrichment of the composition of the kombucha with organic acids and metabolites formed during fermentation and a change in the carbohydrate composition of the drink. Most of the panelists preferred the taste and aftertaste of the traditional kombucha sample K (Figure 7) and liked the KC sample with the addition of only lichen extract the least. Some of the panelists, however, noted a pleasant flavor and curious taste of the samples with the addition of the algae extract (KF) and a mixture of the algae and lichen extracts (KFC).

## 5. Conclusions

In this study, the influence of the addition of extracts of the brown algae *Fucus vesiculosus*, the lichen *Cetraria islandica* and their mixture in the composition of fermented green tea on the microbiological and biochemical properties of kombucha was investigated for the first time. The antioxidant activity of the fermented drinks with the addition of the algae extract (KF), lichen extract (KC) and their mixture (KFC) after 11 days of fermentation was comparable to the antioxidant activity of the control sample K (kombucha). The total content of polyphenols from 4 to 14 days of fermentation in the samples with the addition of the plant extracts was not significantly different than in the control sample. From days 4 to 11 of fermentation, a decrease in the pH of the KF and KFC samples compared to the K and KC samples was observed. In the mature drink obtained by the 12th day of fermentation, the increased content of sucrose and organic acids was detected in the KF sample, while in terms of acetic acid content, the KF and KFC samples showed greater amounts. Fructose and glucose were detected in the largest amounts in the K and KC samples. A decrease in the total amount of yeast was observed in the KF, KC and KFC samples. In addition to the yeast *Brettanomyces bruxellensis* and bacteria *Komagataeibacter rhaeticus* and *Komagataeibacter hansenii* contained in all of the samples, the yeast *Zygosaccharomyces bailii* and bacteria *Komagataeibacter cocois* were also detected in the KF and KFC samples. The highest organoleptic indicators of the parameters “flavor”, “clarity” and “acidity” occurred in the KF sample. The K sample had the highest indicators of the “taste” and “aftertaste” parameters. The results of this study point to the potential of algae and lichens as functional additives to produce non-alcoholic fermented beverages with additional nutraceutical value. More experiments are needed to improve the taste and quality of plant kombucha, including more research on kombucha fermentation kinetics and optimization of the extraction techniques. In addition, it is necessary to carefully study the biological properties of the obtained drinks with plant supplements and prove the advantages and disadvantages of consuming this functional drink.

## Figures and Tables

**Figure 1 foods-12-00047-f001:**
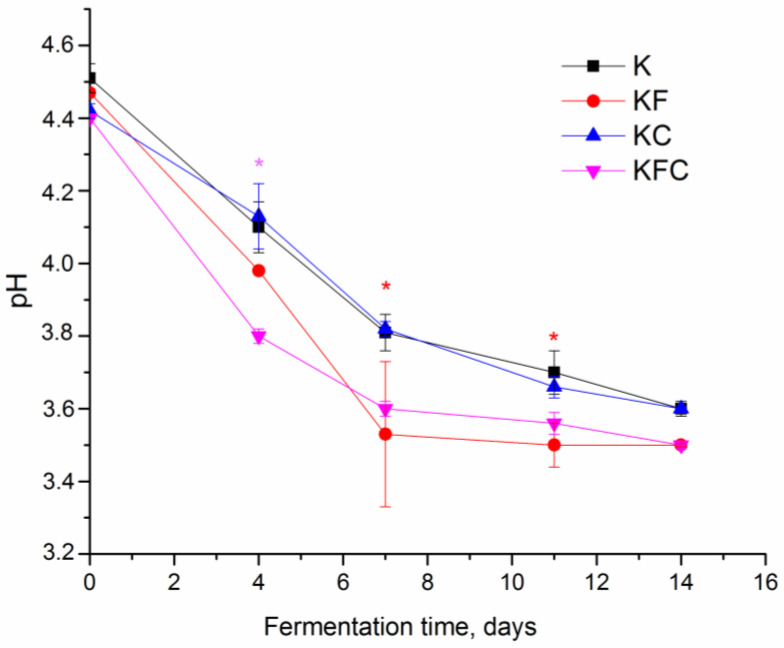
Changes in pH during kombucha fermentation (K—control, KF—kombucha with algae extract, KC—kombucha with lichen extract, KFC—kombucha with a mixture of algae and lichen extracts). * Significantly different from K, *p* < 0.05.

**Figure 2 foods-12-00047-f002:**
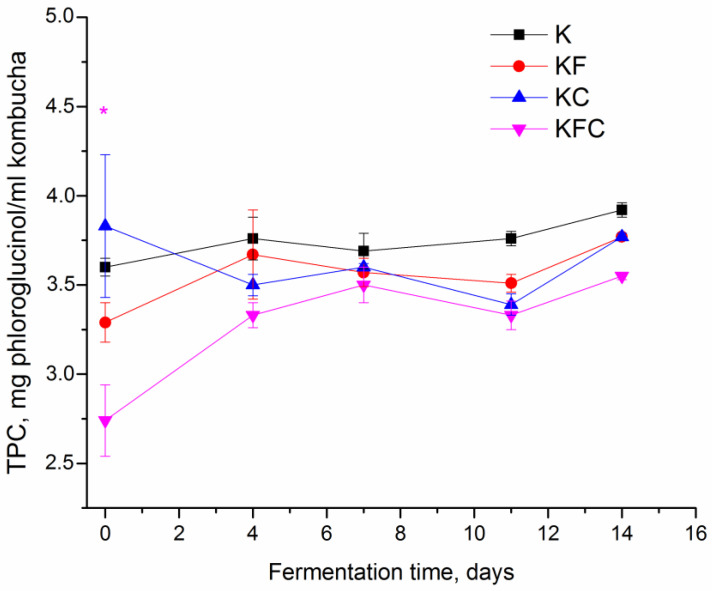
Total polyphenolic content (TPC) of kombucha during fermentation (K—control, KF—kombucha with algae extract, KC—kombucha with lichen extract, KFC—kombucha with a mixture of algae and lichen extracts). * Significantly different from K, *p* < 0.01.

**Figure 3 foods-12-00047-f003:**
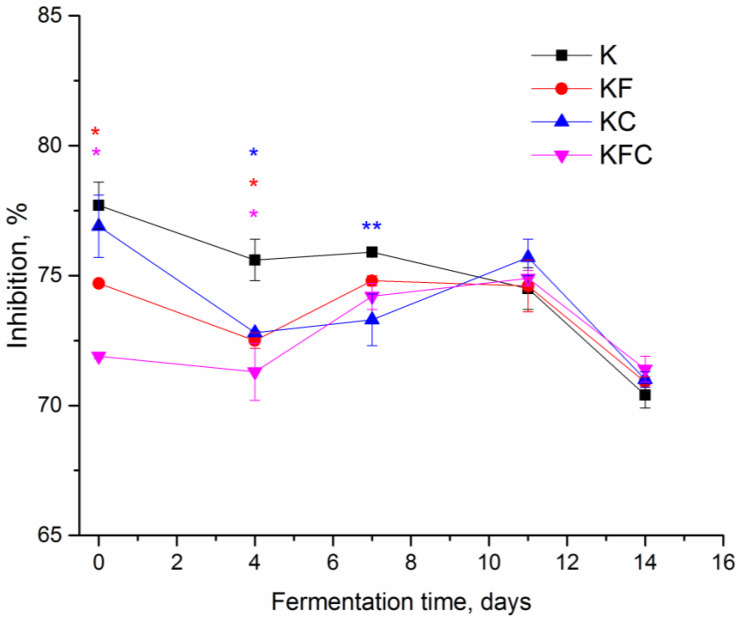
DPPH radical scavenging activity of kombucha during fermentation (K—control, KF—kombucha with algae extract, KC—kombucha with lichen extract, KFC—kombucha with a mixture of algae and lichen extracts). * Significantly different from K, * *p* < 0.01 (** *p* < 0.05).

**Figure 4 foods-12-00047-f004:**
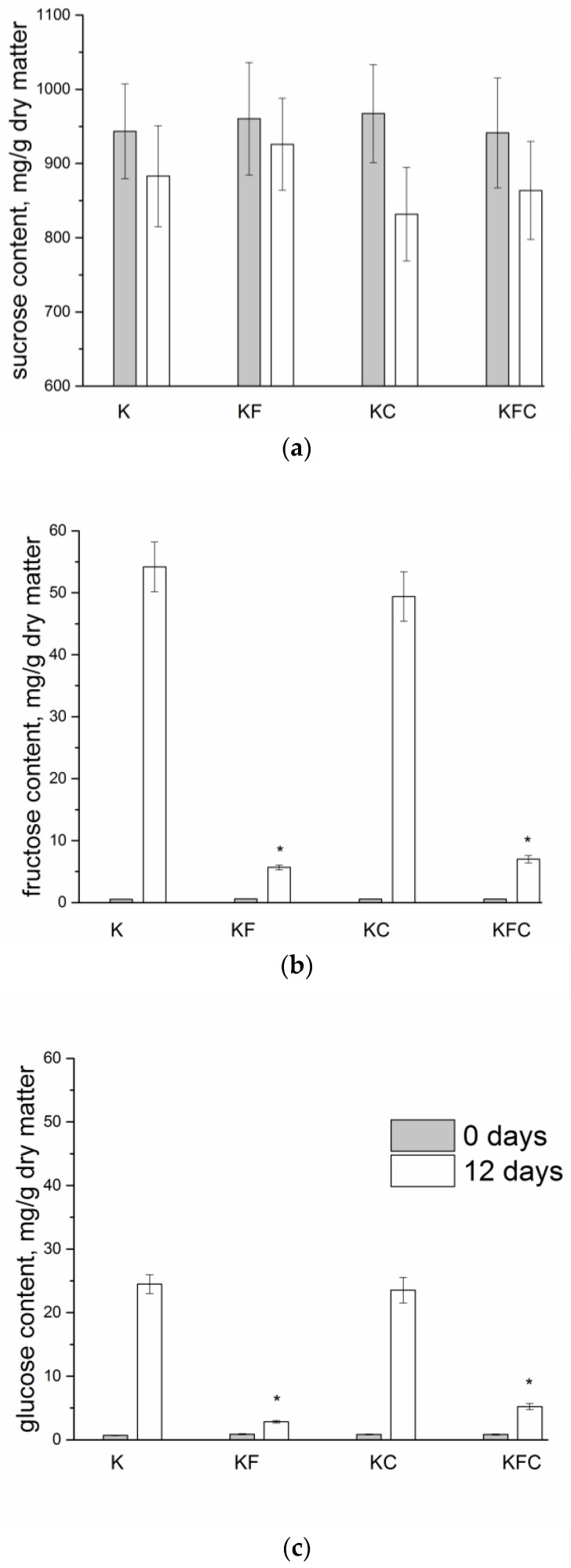
The sugar content of kombucha (K—control, KF—kombucha with algae extract, KC—kombucha with lichen extract, KFC—kombucha with a mixture of algae and lichen extracts): (**a**) sucrose content at 0 and 12 days of fermentation, (**b**) fructose content of kombucha samples at 0 and 12 days of fermentation, (**c**) glucose content of the kombucha samples at 0 and 12 days of fermentation. * Significantly different from sample K, * *p* < 0.05.

**Figure 5 foods-12-00047-f005:**
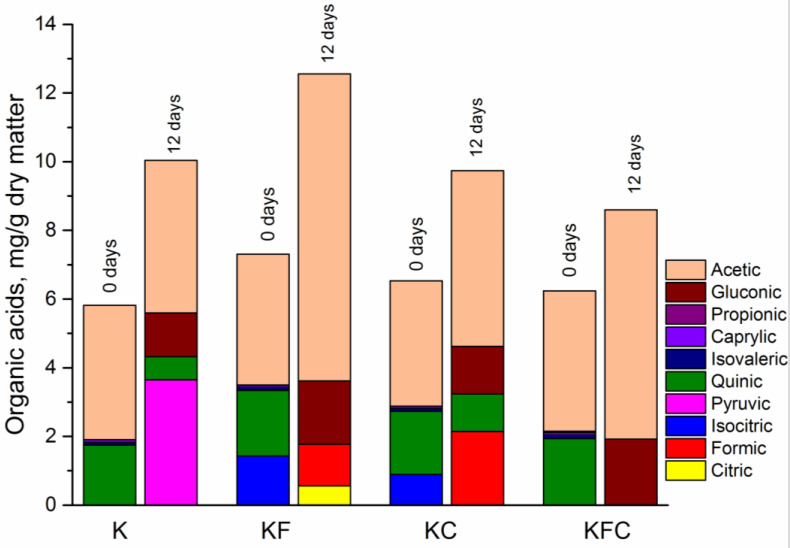
Organic acids content in the kombucha samples at 0 and 12 days of fermentation (K—control, KF—kombucha with algae extract, KC—kombucha with lichen extract, KFC—kombucha with a mixture of algae and lichen extracts).

**Figure 6 foods-12-00047-f006:**
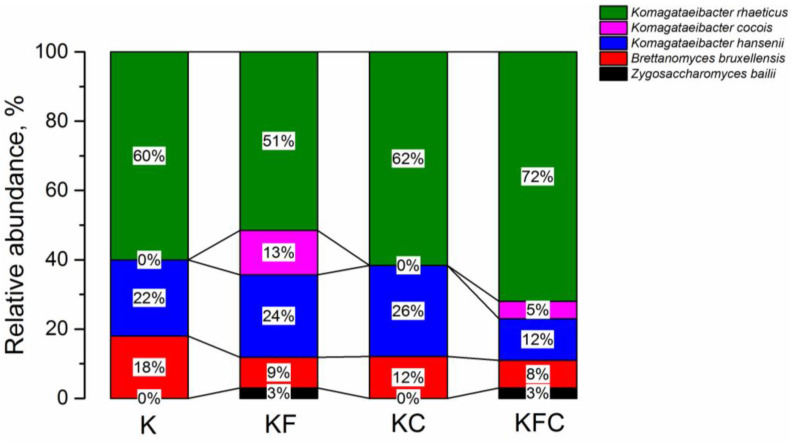
Taxonomic classification of the microbial population of the kombucha samples on the 12th day of fermentation (K—control, KF—kombucha with algae extract, KC—kombucha with lichen extract, KFC—kombucha with a mixture of algae and lichen extracts).

**Figure 7 foods-12-00047-f007:**
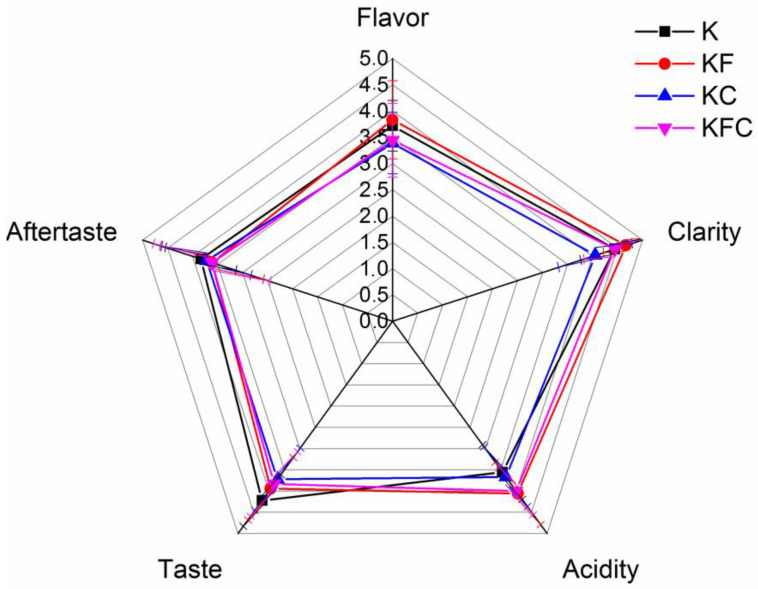
Sensory evaluation of kombucha on the 12th day of fermentation (K—control, KF—kombucha with algae extract, KC—kombucha with lichen extract, KFC—kombucha with a mixture of algae and lichen extracts).

## Data Availability

The data presented in this study are available on request from the corresponding author in accordance with the State regulations and appropriate laws.

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
