# Peer review of "Effect of Brown Algae and Lichen Extracts on the SCOBY Microbiome and Kombucha Properties"

_foods, 2022, doi:10.3390/foods12010047_

Round 1

Reviewer 1 Report

The manuscript "Effect of brown algae and lichen extracts on the SCOBY microbiome and kombucha properties" by Golovkina et al is interesting.

To increase the rigorous and robbust, here are my comments:

1. Add more keywords: nutraceutical, synbiotics; this will improve the SEO and Readability of this manuscript if published.

2. There have been researchers who published the manufacture of kombucha from Algae, and demonstrated its nutraceutical properties:

https://doi.org/10.1016/j.heliyon.2021.e07944; https://doi.org/10.1016/j.clnesp.2022.04.015; https://doi.org/10.1017/S002966512200891; https://doi.org/10.1017/S002966512100272X

It will be more powered if the introduction mentions that the use of algae has also been shown to increase promising bioactive compounds and activities in kombucha products.

3. The subtitles must be in accordance with the writing guidelines of the journal. For example, "Preparation of medium for culturing SCOBY" should be "2.1 Preparation of medium for culturing SCOBY", etc. Check it all out!

4. Details of brown algae and lichen must be given, coordinates of the location (can be found on google maps) where the sample was obtained. Where and by whom is botanical authentication performed? must be detailed.

5. The room temperature must be clearly stated in Celsius.

6. Each method of analysis should be given an appropriate reference. For example, DPPH, whose research are you referring to? or modify study who? reference details must be provided.

7. Authors must pay attention to the writing of international scientific principles. The comma must be written in full stop. For example, line 116 "0,5 and 1,5" should be "0.5 and 1.5". Check and correct throughout the manuscript.

8. Some tests/analysis were carried out after fermentation of kombucha products for 12 days, some were carried out after 11 days. Does this make sense?

For example, chemical analysis is carried out on 12 days of fermentation, but sensory evaluation is carried out after 11 days?

The length/time of fermentation affects the taste and also the components.

9. Line172 should be "Statistical processing analysis".

10. It would be better if the EC/IC50 data from DPPH and TPC were in this manuscript.

11. Reference, give the doi number of each publication that has it.

Reviewer 2 Report

This is a very interesting paper. Although all imperfections of it, I think that this paper can be processed further after minor revision.

These are my suggestions:

- Reorganisation of 2. Materials and Methods are required. Please, divided all subtitles in some manner. For example, 2.1. Chemicals, 2.2. Preparation of experimental set ups (2.2.1. Preparation of medium for culturing SCOBY; 2.2.2. Algae extract preparation), etc.

- Add references for Sensory evaluation as well as all required details about panellists and procedure for tasting

- Figure 4 needs to be larger - all bars and letters (as Figure 5)

- Avoid using "our" in the whole paper; all sentences must be in a passive form;

Round 2

Reviewer 1 Report

In the current version, the manuscript looks more rigorous. I recommend for publication.